# Key Considerations When Providing Physical Rehabilitation for People with Advanced Dementia

**DOI:** 10.3390/ijerph20054197

**Published:** 2023-02-26

**Authors:** Abigail J. Hall, Fay Manning, Victoria Goodwin

**Affiliations:** Department Public Health and Sports Sciences, University of Exeter, Exeter EX1 2LU, UK

**Keywords:** advanced dementia, rehabilitation, physical interventions, qualitative, healthcare professionals

## Abstract

Dementia is a growing global challenge with numbers set to increase rapidly in the coming years. Evidence suggests that exercise can be effective in improving cognitive functioning, but the evidence does not yet support improvements in other key domains such as quality of life or physical ability. The aim of this study was to explore the key components that needed to be considered when providing physical rehabilitation to people with advanced dementia. The study used a qualitative approach involving semi-structured focus groups with health care professionals who are experts in delivering interventions to people with advanced dementia. As a pragmatic study seeking to inform the development of interventions, a thematic coding approach was used to make sense of the data. We collected data from 20 healthcare professionals who reported that key considerations needed to be considered from both an assessment and an intervention perspective. The assessment needed to be person centred and, with the right people engaged and using outcome measures that were meaningful to the patient. The actual intervention also needed to follow the principles of person-centred care, with emphasis placed on the importance of taking time to build a rapport with the person, but also reducing any of the barriers that would prevent effective engagement, such as unsuitable environments. Our study suggests that while there are barriers and challenges to providing interventions and rehabilitation to people with advanced dementia, appropriate person-centred, tailored interventions can be effective and therefore should be offered.

## 1. Introduction

Dementia is an umbrella term used to describe a set of disorders that affect the brain, with over 100 established different types of dementia [1], but it can be broadly categorised into four main types: Alzheimer’s, vascular, Lewy Body and Frontotemporal—although many have mixed aetiologies. Despite having different symptoms and trajectories of disease progression, dementia generally results in a global and continuing loss of cognitive and intellectual functioning, leading to difficulty maintaining social and occupational performance [2].

As a chronic and progressive disease, it is ultimately a fatal neurodegenerative condition [3]. The initial stages of dementia may only present with discrete and almost undetectable symptoms, however, advanced dementia is characterized by profound cognitive impairment, absence of verbal communication and complete functional dependence [3]. People with dementia fall twice as often as their cognitively intact counterparts [4]. Research suggests this is because of increased gait variability and instability [5], and the result is a greater incidence of fractures [6]. 

Advanced dementia is one of the leading causes of death in the UK and United States [7] and typically presents with devastating memory deficits, challenges with communication and a loss of physical ability [7,8]. While there is a gradual and significant loss of physical ability, there is little evidence to determine whether physical interventions can be effective in slowing or reducing this physical decline [9]. 

There is some evidence to suggest that physical activity and exercise are effective strategies in improving cognitive function in older people [10,11], while also improving the ability to perform activities of daily living [11]; however this remains controversial with a more recent study suggesting that exercise had no beneficial effects on cognition [12]. Equally, the evidence supporting physical activity for people with dementia to improve physical functioning remains unclear. A recent scoping review highlighted the paucity of evidence to support interventions for people with advanced dementia [9], with most of the research focusing on people with mild to moderate cognitive difficulties. Aligned with this, there is no qualitative research exploring the experiences of health care professions delivering rehabilitation interventions to this population. Thus, the aim of this study was to explore the important components and considerations when undertaking physical rehabilitation interventions for people with advanced dementia.

## 2. Methods and Design

Semi-structured focus groups were undertaken with a range of health professionals involved in rehabilitation who had specialist skills and knowledge of treating people with dementia. In view of the COVID-19 pandemic and restrictions in travel and meetings, the focus groups were undertaken virtually. This also reduced barriers due to geographical limitations and enabled international experts to contribute. The study has been reported according to the COREQ guidelines (Appendix A).

### 2.1. Setting and Participants

Participants were sought who specialise in working with people with dementia. These included professionals such as physiotherapists, occupational therapists, nurses and researchers. There were no limitations regarding age or geographical location, but the participant needed to be an expert in their field of work, so a purposive sampling strategy was employed. A targeted social media advert sought professionals with advanced skills working with people with advanced dementia and pre-existing networks were drawn upon to recruit participants. Some of these participants were known to the researchers in a professional capacity due to previous work undertaken in this area. This did not affect the data collection. 

#### Inclusion Criteria

Specialists working with people with dementia as a physiotherapist, occupational therapist, psychologist, nurse, or other related profession.

As part of the study, a patient, public, involvement and engagement (PPIE) group was formed. The aim of this group was to ensure that the patient and public voice was not only represented but that the study was developed in collaboration to ensure its relevance. The group consists of a person with dementia, several carers, a retired physiotherapist, and an occupational therapist who is a carer for a person with dementia. Representation from this group was sought for each focus group, however due to practical challenges they were unable to join the focus groups. Therefore, discussion with PPIE representation followed the data collection phase and contributed to the interpretation of results. Nobody other than the participants and the researchers were present during the data collection.

Any participants who responded to the social media advert or email sent to professional networks were sent an email with the participant information sheet and offered an opportunity to discuss the study further. Participants who were contacted directly were sent an introductory email about the study and asked to contact the lead researcher if they were interested in taking part. Potentially interested healthcare professionals (HCPs) were contacted via email and offered an opportunity to have a telephone or Teams/Zoom call to discuss the study in more detail. Participants were provided with a participant information sheet and asked to provide informed consent if they were willing to participate. Participants were informed that they were under no obligation to participate in the focus group and they may withdraw from it at any time, without any negative consequence, up until the point where the data were fully anonymised. 

Provided they met the inclusion criteria, all participants who consented to take part were invited to join a focus group. Recruitment continued until sufficient data was obtained to be able to answer the aim of the study.

### 2.2. Data Collection

Focus groups were undertaken with a range of participants to explore the challenges and techniques required to effectively manage people with more advanced dementia. They were facilitated by an experienced qualitative researcher (AH). A second team member (FM), also an experienced qualitative researcher, acted as a ‘second facilitator/observer’, taking notes and observations pertaining to interaction of the group. Both researchers have undertaken several training courses pertaining to qualitative analysis and have published multiple qualitative papers.

The focus groups were undertaken on Microsoft Teams and recorded to enable ease of transcription and review of data. The focus groups lasted for a maximum of 90 minutes which was the scheduled duration for the meeting. The participants were informed that this study formed part of a larger piece of work aiming to develop interventions for people with advanced dementia. The facilitator utilised a semi-structured guide (Appendix A) where participants were asked the same initial questions and the questions were worded so that responses were open-ended. The topic guide was not piloted prior to the initial focus group as it was anticipated that it could be adapted for subsequent focus groups if needed. This was not needed, and the same topic guide was used for all interviews. This open-endedness allowed the participants to contribute as much detailed information as they desired, and it also allowed the researcher to ask probing questions as a means of follow-up. The focus group questions then varied according to how the group responded and interacted. The transcription function on Teams was used to transcribe the focus groups. This had been tested and demonstrated to be effective and reliable, but the researcher then checked all recordings and transcripts to ensure accuracy. The data was anonymised so that individual participants could not be identified, and all participants consented to their data being transcribed and analysed. Transcripts were not reviewed by participants. The data collection from the focus groups provided rich and detailed data and no further follow-up interviews were deemed necessary.

### 2.3. Analysis 

Analysis began after the first focus group was completed and continued through the following focus groups. As the data collection and initial analysis ran simultaneously, arising themes or gaps could be probed accordingly in the remaining focus groups. 

In line with the aims of this study to determine key considerations when designing a physical rehabilitation intervention, a thematic coding approach [13] was employed as this would facilitate the development of key themes or patterns that emerged from the data. An inductive approach allowed the development of themes that were of relevance rather than testing any existing theory. Familiarization of the data was undertaken immediately following transcription, followed by a process of open coding by one of the researchers (AH). Electronic software (NVivo) was used to manage this coding process. Following this open coding, themes were collated, and core categories identified in a process of selective coding. These themes were then discussed with the PPI representative as well as the other two researchers. Any disagreement in analysis of the data and subsequent generation of themes was resolved by discussion.

The data gathered from the focus groups were analysed to determine the core domains that needed to be considered when developing the intervention. These core domains would then be used during a subsequent expert consensus process.

## 3. Results

Three focus groups were undertaken with a total of 20 health care professionals (Table 1). One person who expressed an interest in taking part did not reply to the meeting invite and therefore did not take part. From those who did agree to take part, nobody withdrew from the study. The healthcare professionals consisted of physiotherapists, occupational therapists, nurses and researchers all who were specialists treating people with dementia with an average of eighteen years of clinical experience and an average of thirteen years specializing specifically with people with dementia. They had experience of working with people with dementia in a variety of settings including in-patient settings, out-patient clinics, patients’ own homes and residential/nursing settings. Most participants were from the UK, with one person working elsewhere in Europe. Approximately half of the participants reported working in mental health settings, with the remainder aligning with physical health services. Following completion of the third focus group it was felt that sufficient data was obtained to answer the aim of the study.

Thematic coding was used to determine the key components of the physical rehabilitation intervention and thus explore the key domains. The data was discussed in terms of assessment and the actual intervention that was delivered as these were the key components discussed.

### 3.1. Assessment

One of the keys to providing successful input for a person with advanced dementia was reported to be the initial assessment. Several key considerations were identified when undertaking an assessment to ensure that it was effective and allowed the HCP to get a true understanding of the person with dementia. The aims of the assessment were to provide the HCP with sufficient background and understanding to effectively deliver their intervention.

### 3.2. Time 

It was often reported that the assessment may need to span over a series of contacts rather than a single contact and this would often take a considerable amount of time. In community settings, it was reported that assessments often could take several hours. The ability to undertake such a long and comprehensive assessment was variable with acute settings reporting that there simply was not sufficient time or resources to be able to offer such extended assessments.


*Yeah, I think that’s a particular issue and we bang on about it in acute, so apologies, but time is so, so limited. It’s limited for us all. But you know when you have a geriatrics ward of 24 patients and 15 of them are on your caseload.… And then you come across the patient with severe or advanced dementia. What they need is time. They really, really do need time. You need time to build a rapport. You need time to get to know them. They need to get to know you.*
(PT2)

It was also noted that the timing (e.g., morning/evening) of the assessment could impact the quality. Therefore, they needed to be undertaken at a time most suitable to the participant and taking into account variable capacities throughout the day. This was easier for HCPs who were based on the ward or care home where the person was currently residing, however for community-based HCPs, it was difficult to ensure they visited the person at the most appropriate time.


*…not just go in the morning and think that’s it. I can’t go back to them, but actually that persons not engaged at that point back and go back later in the day or later in the day and having that time to maybe invite family and for sessions or carers that know that person really well.*
(PT10)

### 3.3. Involving Others

It was also reported to be important to have the ‘right people’ present during the assessment. This may be a family member or carer if this was appropriate, but in other situations it was reported that having a family member present may be distracting and not add value to the assessment. Carer stress was reported to often be high and understanding the pressures that the carers often faced was vital as this would inform whether they might add value to the assessment. 


*I spend a lot of my time speaking to family members caring for someone living with dementia at home, and it’s always so striking that they’re kind of unmet needs as a carer directly impact on the care they’re able to provide for the person they’re caring for and the lack of support.*
(PT1)

Where a person was in a care home or nursing home, it was suggested to be valuable to have one of the carers present to help ensure the background of the person was better understood. Where possible, it was valuable for the person to have an allocated “key worker” who could provide consistency of approach and communication for the person with dementia.

Overall, it was noted that it was vital to ensure effective multidisciplinary team working with all those involved in the care of the person with dementia to ensure a seamless approach to care that is person centred.


*…our strength perhaps lies in our MDT and the different skills that everybody can bring to make that person’s care so person centred to them.*
(RN2)

### 3.4. Type of Assessment

The content of the assessment needed to focus on what was important to the person with dementia and explore their ability to undertake meaningful activities. HCPs reported that typically assessments needed to be holistic for this patient group, exploring co-morbidities and concurrent medical difficulties that might be affecting the person’s physical abilities. 


*And one of the things I’ve loved about being in advanced clinical practitioner is it’s enabled me to take on the skills that I that I became so increasingly frustrated with that I couldn’t deal with. I couldn’t deal with constipation, urinary tract infections, chest infections, which are what we need for our holistic assessment, isn’t it?*
(PT1)

Many references were made to the value of using the “This is Me” guidance document [14], which is a tool created by the Alzheimer’s Society UK to support person centred care for people with dementia. The document, or similar, was commonly seen in acute or residential/nursing home settings but was reported to be less commonly utilised for people living in their own homes, although there was no evidence to make sense of why it was used less in people’s own homes. The tool was useful to ensure that the HCP had a better understanding of the person before undertaking their assessment, including having an accurate idea of their baseline functioning which was a key consideration during their assessment. 

### 3.5. Outcome Measures

A part of the assessment was the use of outcome measures. It was felt that outcome measures were important to prove effectiveness of interventions—particularly for this population where people with advanced dementia were commonly reported not to have “rehabilitation potential”. However, the use of outcome measures was challenging for HCPs. It was reported that ‘traditional’ outcome measures weren’t appropriate for this population.


*Because we’re using outcome measures that were developed for very quantitative outcomes within MSK and I guess to a little certain extent towards the very precise area of stroke rehab.*
(PT1)

Therefore, outcome measures needed to be tailored to suit the needs of the person with dementia and be meaningful to that person. In the acute setting it was reported that outcome measures were particularly challenging due to the rapid flow of patients on and off the ward. Furthermore, outcome measures were reported to historically be used to demonstrate positive improvements; however, patients with advanced dementia have a deteriorating chronic condition and thus outcome measures should reflect that slowing deterioration is a positive outcome. Goal setting and outcome measures were closely linked, with achievements of goals being termed a positive outcome and thus measuring outcome against goals was suggested to be a more effective way of determining outcomes.


*And we hear a lot about person centred care and what matters to me, and all of these things. But unless we’re actually start to embrace those as outcomes and actually use those as proof of that, what we do works, they’re never going to be imbedded in in traditional approaches to rehabilitation.*
(R1)

### 3.6. The Intervention

There were several elements to consider when determining the best way to deliver an intervention to the person with advanced dementia.

### 3.7. Approach

Our participants unanimously reported that interventions and approaches needed to adhere to the principles of person-centred care and thus, all interventions needed to be specifically tailored to the individual. The role of the HCP was described as being to *“facilitate movement rather than request movement” (PT9)* and thus the aim of the intervention was to determine the best approach to achieve this facilitation. This involved an investment of time to get to know the person to understand how they could best engage them. For people with advanced dementia, the level of physical ability may be very varied and therefore, the actual intervention offered would have to consider the level of physical ability as well as all other factors. Interventions were reported to need to be meaningful to the participant with examples given of doing the weeding in the garden or going to the boxing gym. This meaningful activity was related to everybody, not just those with advanced dementia.


*It doesn’t really matter if a patient has advanced dementia or if they’re me, a 28-year-old that’s had knee surgery. If an exercise makes no sense to me, or if it’s boring, or if it’s not gonna help me get back to what I want to do, I’m probably not gonna do it. And. And really, if we make healthcare better for those with advanced dementia, we’re gonna make it better for everyone, because what we’re talking about there is personalized care, isn’t it?*
(PT4)

Functional activities were seen as key rather than using specific exercises which would be unlikely to be understood or engaged with. This focused on promoting and encouraging mobility or transfers or trying to challenge a person’s balance during meaningful activity. Meaningful activities were discussed at length. While tools such as music were discussed to be useful to help engage with a person, this music needed to be something that the person enjoyed—thus highlighting the importance of taking time to get to know the person.

### 3.8. Building Rapport

There was a real emphasis that part of the intervention itself required time and effort to get to know the person and build rapport with them. Interestingly, some participants described this as a “non-traditional” approach and when questioned, reported that this element of the contact often went unnoticed and would not be documented as part of the contact.


*…it is that that time spent with the rapport building, it’s the cups of tea and it’s the, you know, sitting picking at weeds in the garden and you know, and that much more sort of functional and pragmatic approach to things, and I think that as a student, if you don’t understand that that’s actually valuable.*
(PT8)

Gaining the person’s trust was deemed to be vital and this was all part of building a rapport with the person. It was reported that often as HCPs we ask the person to do something that they are fearful of or causes them pain. With the challenges of not being able to explain due to their cognition, this ability to build rapport also allowed the development of trust to build and therefore would be more likely to be able to engage with the person.


*The more I gain the trust, the more they’re able to engage with the therapy and do along with them.*
(PT11)

As part of this building rapport with a person, it was deemed vital to ensure continuity of the HCP seeing them. While this was possible in mental health and community settings, it was more challenging in acute settings where staffing and resources were focused more on the flow of patients through the acute setting and facilitating discharge. HCPs recognised the importance of continuity and attempted to provide continuity as much as possible, but within the restrictions of the services they worked within.

Where it was not possible for the same HCP to provide input to the person, the value of effective communication within the MDT was highlighted. In care homes, this involved engaging with the formal carers to ensure that they were supporting the person with dementia in a way that promoted the intervention of the HCP. Ideally, the carers who knew the person well were encouraged to actively participate in the HCPs sessions with the person with dementia to ensure continuity. Having a person who knew the patient well allowed the rapport to develop more easily and rapidly. This was reported to be ideal, but often was not possible. 

### 3.9. Reducing Barriers

In order to effectively work with a person with advanced dementia it was reported to be vital that any negative barriers that would impact on the intervention. The main barrier to treatments was reported to be the presence and poor management of pain. Ineffective pain management was cited to be the biggest barrier to providing any interventions.


*…there’s not always a recognition of basic care needs for people with advanced dementia. So someone who’s distressed, agitated act now, you know, behaving in a certain way, she’s right. It’s all blamed on “ah well, they’ve got dementia”. And actually sometimes it is just a basic care need that hasn’t been met like pain. Pain is one of the biggest underrecognized things*
(PT7)

The environment was also seen as often being a significant barrier to effective interventions. Our participants described a variety of challenges that the environment caused—in community and acute settings. There was a common feeling that the acute environment was not the most appropriate place to provide interventions for people with advanced dementia. However, care homes, or a person’s own home came with their challenges too. The acute setting was reported to be a challenge due to the amount of noise and distractions that were often present in a busy ward setting.


*You know, there can be behind a curtain, but next door there’s somebody talking to another patient trying to get them on a commode, giving them instructions, and then they’re wanting to follow them or they wanting to listen to something else.*
(PT13)

While it was noted that it was ideal to have a quiet space away from the ward where there were fewer distractions, this was not always available and when it was, it required sufficient time to be able to take the person there.

While it was suggested that care homes or nursing homes would provide a better space for a person to receive interventions, these were fraught with challenges such as getting the right equipment and ensuring the care staff had sufficient skills and experience to manage the person while also promoting activity and movement. Our physiotherapists reported that it was often their role to educate the carers to ensure that they were able to meet the physical needs of the person. Lack of knowledge of dementia was reported as being common, so this was a significant barrier that HCPs felt was part of their remit to provide this education.

Co-morbidities were also seen as a potential barrier to providing an intervention or rehabilitation to a person with advanced dementia. It was seen as key to consider the person from a holistic perspective and gain a deep understanding of the presence of co-morbidities, but also how these other conditions might affect the person’s ability to engage with an intervention.

## 4. Discussion

The aim of this study was to explore the important components of interventions and considerations when undertaking physical rehabilitation interventions for people with advanced dementia. There is currently little evidence to determine the effectiveness of physical interventions for people with advanced dementia [9], with the majority of studies exploring interventions for people with less severe dementia, or focusing more on cognitive rather than physical rehabilitation.

Our data highlighted the importance of delivering person-centred care (PCC) to people with advanced dementia. The concept of PCC originates from the work of Carl Rogers [15] and in people with dementia by [16,17]. The principle is that the person is placed at the centre of their own care. Our participants reported the challenges of providing PCC in various settings, especially in acute settings where the priorities were to facilitate discharge. Several authors have explored the challenges of providing PCC in acute settings and found that, like our participants, there were examples of good practice, but there were further opportunities to facilitate personhood that HCPs did not take such as taking the person to a more meaningful place—such as a gym—to deliver their rehabilitation, citing the lack of value placed on PCC by the organisation [18,19,20] and therefore the time and resources that they were able to make use of.

However, PCC went beyond people with dementia. Our participants highlighted that all care should be person centred—whether the person had dementia or not—and that health and care services needed to focus on the person design services and interventions around them—rather than expecting the person to adapt to the already existing services. This is fundamental to supporting an individual’s Human Rights and aligns with the World Health Organisation’s “*Global Strategy on People-centred and Integrated Health Services*” [21].

It was also highlighted that the assessment—including outcome measures—needed to be person centred and apply a biopsychosocial approach. It was noted that outcome measures often were not suitable for a person with advanced dementia where they comprised of physical or biomedical domains. Systems and organisations appear to drive the need to prove effectiveness of physical intervention with outcomes that align with physical measures of ability, with less reliance places on outcome measures that evaluate quality of life. Spector and colleagues [22] report the lack of knowledge of outcome measures for people with dementia and emphasise the need to develop more “robust, contemporary measures of knowledge”.

Our participants reported many challenges that they faced when treating people with advanced dementia and thus, the key was to facilitate the best environment and opportunity for the person to improve as possible. There has been a plethora of research exploring the most suitable environments for people with dementia [23,24,25] which has all highlighted it being a key consideration to managing a person with dementia. Our participants reported that this involved removing any potential barriers that might influence the persons’ ability to engage and benefit from the intervention. These were often practical challenges such as noise, distractions or being in an unfamiliar environment. The ability to overcome these barriers varied amongst our participants and settings—with many reporting that they had to “make the best of what they had available”.

## 5. Conclusions

While there are barriers and challenges to providing interventions and rehabilitation to people with advanced dementia, appropriate person-centred, tailored interventions are reported to be effective for this population. Our study sought to pragmatically explore some of these challenges as well as how interventions could be adapted to be effective. Healthcare systems are typically not designed to support this approach to care, focussing on the needs of the system rather than the individual. This leads to HCPs needing to find approaches to overcome these challenges to ensure that they provide the best interventions they can for their patients. The system’s over-reliance on biomedical approaches to measuring outcome means that their efforts are not always visible or rewarded. Research is needed to develop and evaluate physical interventions that are specifically designed for people with advanced dementia.

## Figures and Tables

**Table 1 ijerph-20-04197-t001:** Characteristics of participants (*n* = 20).

Sex	Female	19
	Male	1
Profession	Physiotherapist	14
	Occupational Therapist	3
	Nurse	2
	Other	1
NHS Grade	6	6
	7	8
	8+	2
	N/A	4
Years of clinical experience	0–5	1
	6–10	2
	11–15	1
	16–20	3
	21–25	6
	26–30	2
	30+	3
	N/A	2
Years of experience working with people with advanced dementia		
	0–5	4
	6–10	2
	11–15	6
	16–20	5
	21–25	2
	26–30	1
	30+	0
Location	England	13
	Wales	2
	Scotland	4
	Denmark	1
% of caseload of people with dementia		
	0–25	1
	26–50	1
	51–75	6
	76–100	8
	N/A	4
Locations of work	In-patient	12
	Out-patient	4
	Care home/residential homes	11
	Own homes	11
	Other	2

## Data Availability

The datasets generated during and/or analysed during the current study are available from the corresponding author upon reasonable request.

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
