# Peer review of "Key Considerations When Providing Physical Rehabilitation for People with Advanced Dementia"

_ijerph, 2023, doi:10.3390/ijerph20054197_

Round 1

Reviewer 1 Report

This manuscript reported the interventions and considerations when undertaking physical rehabilitation interventions for people with advanced dementia. This work is interesting and well organized with proper novelty.

1.     Introduction. Please add more information about related work previously published.

2.     Table 1. Female and male were not the equal. Then, how to justify the result unbiased?

3.     Reference. Please add page numbers in ref. 1.

Author Response

Many thanks to the reviewer for taking the time to comment on this paper. We have addressed your comments as follows;

  • we have added additional text to explain the current existing (or in this case, the absence of literature) in this field. Hopefully this justifies the purpose of the paper
  • indeed the majority of our participants were female. We have added text to describe this. We feel this reflects the proportion of HCPs who specialise in this area of work, nor do we feel that the sex of the HCP will unduly influence the results as this is a qualitative piece of work
  • the page number has been added - many thanks for noticing this

Reviewer 2 Report

The paper presents a study for determining the key components that needed to be considered when providing physical rehabilitation to people with advanced dementia. The paper  used a qualitative approach involving semi-structured focus groups with health care professionals who are experts in delivering interventions to people with advanced dementia. Here are my main comments for the paper:

- Thre originality of the paper does not seem to high. What brings new the paper it is not clear

- I the light of the previous comment, a part with related work and how the current paper advance the knowledge and bring originality is missing.

Given the two comments above I will give a major revission

Author Response

Thank you to the reviewer for taking the time to review this paper and suggest some improvements. In line with your comments, additional text has been added in the introduction to explore the existing (or lack of) evidence in this area which helps to  justify the purpose of this paper.

People with advanced dementia are often excluded from research and therefore it is key to understand the challenges that health care professionals may face when treating this population and how they can be overcome. The novelty is the exploration of advanced dementia which our research has shown is entirely different to treating people with more mild forms.

Many thanks

Round 2

Reviewer 2 Report

Thanks authors for their reply. In the reply reasons for novelty is given, but still I miss in the article sufficient literature review on related work and some clear sentences regarding novelty of the work. I would suggest again this to be added more substantially. 

I will give a minor revision

Author Response

Dear reviewer,

Many thanks for your review. I note that you state that everything "must be improved", however your comment only relates to the introduction. I have, however, added some further text to the introduction to hopefully explain the novelty of the article. We thank you for your time and comments.

Many thanks.